# WHAT CAN YOU DO WHEN YOU HAVE ZERO REWARDS DURING RL?

## ABSTRACT

Reinforcement learning (RL) with outcome-based rewards has proven effective for improving large language models (LLMs) on complex reasoning tasks. However, its success often depends on the base model occasionally sampling correct solutions. When no correct solutions are sampled, training encounters a zero-reward barrier where learning stalls due to zero gradients. We study this scenario through the graph search task introduced in Bachmann and Nagarajan (2024) and evaluate recent methods that incorporate desirable components such as dense rewards, diversity incentives, and improved credit assignment. Our experiments show that none of these approaches overcome the zero-reward barrier if the base model never produces a correct answer. In contrast, we find that a simple data-centric intervention of adding easier samples to the training set enables the model to eventually solve the original hard task despite starting from zero reward. Importantly, this succeeds without modifying the RL algorithm itself. Because official implementations of several baselines were unavailable, we developed our own, which allowed us to conduct a detailed analysis of their failure modes. We release these implementations to support further research: https://github.com/anon-zero-rewards/zero-rewards-rl.

## 1 INTRODUCTION

Reinforcement learning (RL) has emerged as a crucial tool in the post-training of large language models (LLMs) for tackling complex reasoning tasks such as mathematical problem solving (Guo et al., 2025), web navigation (Putta et al., 2024), and algorithmic discovery (Fawzi et al., 2022). These advances often rely on sparse rewards, where the model receives only a binary correct-or-incorrect signal at the end of its response. Although such outcome-based rewards can significantly enhance model accuracy, their effectiveness typically relies on starting from a reasonably strong base model (Yue et al., 2025; Gandhi et al., 2025).

When the base model fails to solve a task even after repeated attempts, RL training becomes ineffective since zero rewards yield zero gradients, leaving the model parameters unchanged throughout training, preventing further scaling of RL.

This begs the question:

> *What can one do if there are zero rewards due to no correct answers*
> *being sampled by the model during RL post-training?*

To study this question, we consider the simple task of searching for a path from a source node to a target node in a star graph from Bachmann and Nagarajan (2024) (see Fig.1). As we will discuss later, this task enables a controlled and systematic study of how different RL post-training methods perform under zero outcome rewards (see Sec.2).

While one could address the issue of zero rewards by performing supervised fine-tuning (SFT) on human-written or model-generated traces and then apply RL (Gandhi et al., 2025), *we deliberately rule out this option for the sake of this study*. This lets us isolate and evaluate methods specifically intended for scenarios with zero outcome rewards during RL training.

The above assumption leaves us with several interesting approaches to tackle the problem of zero rewards. For instance, one could apply *reward shaping* (Setlur et al., 2025a; Qu et al., 2025) to

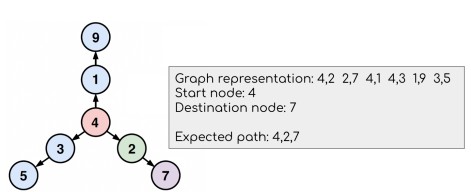 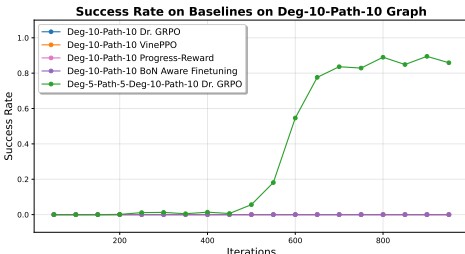

Figure 1: **(Left)** Illustration of the `Degree-3-Path-3` task, where the center node (4) has degree 3 and each outgoing path has length 3. The graph is represented as an adjacency list, and given a source node (4) and a destination node (7), the task is to output a path from source to destination (e.g., 4, 2, 7). See Appendix C for the prompt used. **(Right)** Success rates of different baselines: `Dr.GRPO`, `VinePPO`, `Progress Rewards`, and `Best-of-N` aware finetuning, compared with our data-mixing approach, which augments the training dataset with an equal proportion of samples from the easier `Deg-5-Path-5` dataset. The baselines fail to break the zero-reward barrier, yielding zero success on the test set, whereas mixing in easier samples if effective with outcome rewards.

obtain *dense rewards* that provide a learning signal based on the quality of intermediate steps, even when outcome rewards are zero. One could also explore approaches that improve *credit assignment* for intermediate steps in reasoning (Kazemnejad et al., 2024). Alternatively, the model could be incentivized to sample *diverse* responses during RL fine-tuning (Chow et al., 2025), in the hope that at least one response receives a non-zero reward, thereby kick-starting RL training.

However, to our surprise, we find that **all of these methods fail** (Figure 1) on the simple graph search task[1], despite being specifically designed for cases with zero outcome rewards (see Sec. 3). We investigate these methods and discuss possible explanations for their failure (Sec. 4.1 and App. A). Notably, no official code was available for several of these baselines, so we implemented our own versions, which we release for the community to build upon.

In contrast, we show that a simple **data-centric intervention unlocks RL training**. By adding *easier*[2] instances to the training dataset, the model gradually learns to solve the harder task using **outcome rewards only, even when it was initially *unable* to do so**. Crucially, this is achieved **without modifying the RL algorithm at all**.

This strategy can be seen as a form of implicit curriculum (Stojanovski et al., 2025; Setlur et al., 2025b). In the discussion, we explain how this intervention can aid *skill* learning (Eysenbach, 2025), and how *correlated actions* learned from easier samples (see Sec. 5.3) can transfer to harder tasks that the base model could not previously solve.

Our contributions are three fold: **(i):** We benchmark recently proposed RL algorithms that aim to (a) improve diversity in samples (Chow et al., 2025), (b) perform credit assignment (Kazemnejad et al., 2024), or (c) apply reward shaping (Setlur et al., 2025a), and observe that none of these approaches are effective under zero outcome rewards on a graph search task. **(ii):** We demonstrate a simple data-centric approach is able to unlock RL training, where mixing samples of *easier* difficulty helps the model learn to solve the original hard task, even when the model initially could not do so. **(iii):** In the process of benchmarking different approaches, we implement baselines for which no official code was available, and provide a detailed analysis of why these methods are ineffective in zero outcome rewards scenario.

## 2 EXPERIMENTAL SETUP

In this section, we describe the task and outline the baselines we compare against.

**Q: What is the task?**

---

[1]In scenarios where the base model cannot sample a correct answer
[2]Samples where the base model can generate a correct answer and receive a non-zero reward

**A:** We study the graph search problem introduced in Bachmann and Nagarajan (2024). As shown in Fig. 1, the input is an adjacency list representing a star graph, along with a source and destination node. The task is to output the path from the source to the destination, where the source is always the center of the star and the destination is one of its leaf nodes. To benchmark models in the zero-reward setting, we use the `Degree-10-Path-10` graph, where the center node has degree 10 and each branch has length 10. As illustrated in Fig. 1, existing methods fail on this task.

**Q: Why bother about a graph search problem?**

**A:** Although the task may seem simple, it has several properties that make it well-suited for our study: **(i) Controlled difficulty:** The task supports automatic dataset generation at varying difficulty levels, allowing a systematic study of how training on easier instances transfers to harder ones. **(ii) Challenging for transformers:** Prior work shows that transformers struggle to directly output the correct path, but if they perform intermediate reasoning in a chain of thought, they can effectively search through the graph in-context. **(iii) Low reliance on world knowledge:** Solving these tasks does not require external knowledge. This is a critical feature, as it allows us to isolate the core technical challenge, the zero-reward barrier, without the confounding factors of external information such as knowledge of theorems or lemmas which are often useful for mathematical reasoning problems. This also enables rigorous comparisons of methods with modest compute using relatively small models (e.g., 1.5B parameter scale).

**Q: What baselines do we evaluate?**

**A:** We evaluate three recent methods that tackle different challenges in reinforcement learning for reasoning tasks. **VinePPO** (Kazemnejad et al., 2024) improves step-level credit assignment by measuring the change in value between states before and after each reasoning step using Monte Carlo rollouts. Although this slows individual training iterations, it reduces gradient variance and stabilizes learning. Along similar lines, **Rewarding Progress** (Setlur et al., 2025a) combines outcome rewards with step-level advantages estimated under a different policy, encouraging exploration and making it particularly effective on hard problems. **Best-of-N-aware (BoN) finetuning** (Chow et al., 2025) takes a complementary approach by modifying the training objective to better match inference-time goals: instead of requiring all $N$ generations to be correct, it encourages producing at least one correct output among $N$ attempts. This promotes diversity in the model's outputs and increases the likelihood of obtaining a non-zero reward to initiate RL training. Together, these methods represent state-of-the-art strategies for addressing sparse rewards. For details on the objective and implementation refer Appendix A.

**Q: What is the training setup?**

All experiments were conducted on 4 NVIDIA H100 GPUs using Qwen2.5-1.5B-Instruct as the base model. Each experiment was limited to a maximum of 24 hours or 1,000 RL iterations, whichever occurred first. While we used a single model across all experiments, we expect our findings to generalize to other models. The primary difference would be which tasks are challenging: for Qwen2.5-1.5B-Instruct, the `Degree-10-Path-10` dataset is difficult, whereas other models might find it easier or harder. Nevertheless, increasing task complexity would likely lead to failures even for larger models.

## 3 BASELINES FAIL UNDER ZERO OUTCOME REWARDS

Surprisingly, all the baseline methods we tried—namely, naive RL (Dr. GRPO) Shao et al. (2024), VinePPO Kazemnejad et al. (2024), Rewarding Progress Setlur et al. (2025a), and Best-of-N-aware finetuning Chow et al. (2025)—failed on the simple star graph search task, specifically on the `Degree-10-Path-10` instance (see Fig. 1).

While naive RL is expected to fail under zero-outcome rewards, it is notable that the other baseline methods also fail to solve the task, despite being designed to operate under such conditions.

A keen reader may already have several questions about this result. Before proceeding further, we first address these potential questions.

**Q: Is this task even solvable?**

We will see in Section 5 that the `Degree-10-Path-10` task is indeed solvable. Furthermore, Fig. 2 shows that all baselines can solve the *easier* `Degree-3-Path-3` task, in which the center node has a degree of 3 and each outgoing path consists of 3 nodes.

**Q: Are you sure this isn't a hyper-parameter issue in the baselines?**

To rule out any implementation issues, we focused on several important aspects of each method. For `VinePPO`, we increased the number of rollouts used to estimate the value of intermediate states, giving the model additional chances to reach the correct answer. To meet the requirements of desirable provers outlined in Setlur et al. (2025a), we experimented with more powerful provers that can occasionally solve the `Degree-10-Path-10` task. For `Best-of-N` aware finetuning, we followed the recommended practices described in Chow et al. (2025), testing two different schedules for the KL coefficient and increasing clipping for the sample-dependent weights.

Unfortunately, none of our interventions succeeded. A discussion of these experiments is provided in Section 4.1. In Sec. 5 we discuss a simple data-mixing strategy that helps the unlock RL training.

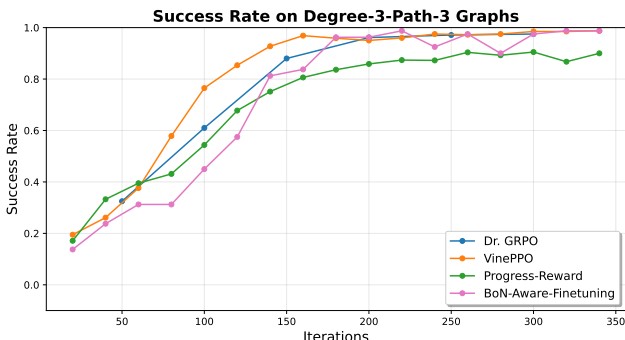

Figure 2: Success rates of different RL algorithms (`Dr.GRPO`, `VinePPO`, `Progress Rewards`, and `Best-of-N` aware finetuning) on a held-out test set of `Degree-3-Path-3 graphs`. These models were trained on `Degree-3-Path-3 graphs`. All algorithms are able to solve the task when the model starts with a reasonable success rate. Furthermore, `VinePPO` converges in fewer iterations compared to `Dr.GRPO`, consistent with findings reported in the literature.

## 4   FAILURE ANALYSIS OF BASELINES

All four baselines, including `Dr.GRPO`, `Progress Rewards`, `VinePPO`, and `Best-of-N` aware finetuning, receive no rewards during training, and thus have zero success rates at test time (see Fig. 1). For the `VinePPO` and `Progress Rewards` baselines, we use Monte Carlo rollouts to estimate the value of intermediate states and compute step-level advantages (refer Appendix A), which makes each RL iteration roughly five times slower. Nevertheless, we continued training these two methods for approximately five times longer than `Dr.GRPO` and `Best-of-N` aware finetuningto rule out the possibility that they might start solving the task at later stages. Despite this extended training, we observe that these methods remain unable to solve the task.

As we discuss in Section 4.1, **a key reason for the failure of some baselines on the `Degree-10-Path-10` task is the base model's inability to occasionally sample correct trajectories.** To rule out any implementation issues, we also experimented with a variant of the task where the center node has a degree of three and each outgoing path has three nodes. In this setting, the base model has an initial success rate of $\sim 20\%$, analogous to the conditions explored in prior work where the base model starts with a reasonable success rate. As shown in Fig. 2, we observe that (i) all algorithms are able to solve the task when the base model begins with a reasonable success rate, and (ii) although each iteration of `VinePPO` is more expensive, it converges in fewer iterations compared to `Dr.GRPO`, consistent with findings reported in the literature.

### 4.1   CASE-BY-CASE ANALYSIS

Here we discuss some of the reasons why the baselines are ineffective on harder variants of the task.

**Dense rewards are not really dense in `VinePPO` and `Progress Rewards`:** Methods like `VinePPO` and `Progress Rewards` go beyond `Dr.GRPO` by computing step-level advantages. However, non-zero step-level advantages are obtained only when there is a change in the value of the state before and after taking the step. This means that for `VinePPO` to produce a non-zero advantage for a step ($\hat{A}^{\pi_\theta}_{y_{c_i}} \neq 0$ in equation 4 ), some of the rollouts under the current policy must succeed. Similarly, for the `Progress Rewards`, some of the rollouts under the prover policy must succeed ($\hat{A}^{\mu}_{y_{c_i}} \neq 0$ in equation 6). In our setting, we observe that throughout training, neither the current policy nor the policy generates a successful rollout. Thus step-level advantages offer no learning signal.

**Instantiating a helpful prover to get a meaningful `Progress Rewards` is hard:** The `Progress Rewards` (Setlur et al., 2025a) work notes that a prover that is too strong or too weak is ineffective: a strong prover cannot distinguish between good and bad steps, while a weak prover fails from most intermediate states, resulting in zero step-level advantages and no learning. Consequently, they identify two desirable properties for provers: (i) the prover should neither be too strong nor too weak, and (ii) it should be reasonably aligned with the policy being optimized.

To satisfy these requirements for the `Degree-10-Path-10` graph, we experiment with two provers. The first, $\pi_{5x5}$, is model trained using `Dr.GRPO` on the `Degree-5-Path-5` task and achieves around $65\%$ accuracy on `Degree-10-Path-10`, partially satisfying the first property. The second, $\pi_{5x5\ \text{mixed with 10x10}}$, is trained using `Dr.GRPO` on an equal mixture of `Degree-5-Path-5` and `Degree-10-Path-10` graphs and reaches around $85\%$ accuracy on `Degree-10-Path-10`.

As shown in Fig. 3, both provers have a reasonable success rate on the `Degree-10-Path-10` task from the start, giving non-zero step-level advantages early in training. The $\pi_{5x5}$ prover provides non-zero step-level advantages about $50\%$ of the time, while the $\pi_{5x5\ \text{mixed with 10x10}}$ prover does so about $60\%$ of the time. However, as seen in Fig. 3, this signal does not lead to better task performance. In both cases, the model responses often become degenerate, repeating characters or words to fill the context window. We believe this happens because the prover policy is not well aligned with the policy being optimized.

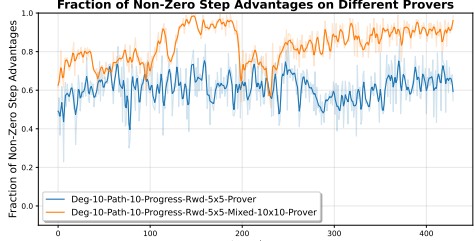 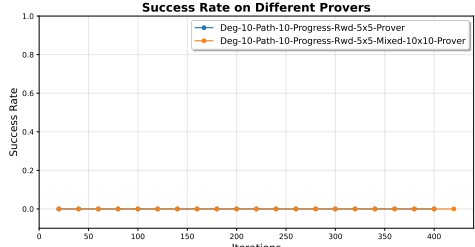

Figure 3: Effect of `Progress Rewards` using different prover policies. **(Left)**: Fraction of non-zero step advantages ($\hat{A}^{\mu}_{y_{c_i}} \neq 0$ in Equation 6) for two provers: $\mu = \texttt{Best-of-4}(\pi_{5x5})$ and $\mu = \texttt{Best-of-4}(\pi_{5x5\text{-mixed-with-10x10}})$, where the models were trained on `Deg-5-Path-5` alone or mixed with `Deg-10-Path-10`, respectively. Both models provide non-zero step advantages for `Progress Rewards` due to their reasonable success rates on the harder task. **(Right)**: Success rate on a held-out test set of `Degree-10-Path-10` examples. Despite using the same two provers, both models fail on the `Degree-10-Path-10` task. We believe this is because the prover policy is not well aligned with the policy being optimized.

**Unstable training in Best-of-N aware finetuning:** We followed the practices suggested in Chow et al. (2025), including (a) using a KL schedule and (b) clipping the sample-dependent weights multiplied by the log probability (Eq. 9 in Chow et al. (2025)). Notably, the KL schedule in Chow et al. (2025) is quite aggressive, starting with a coefficient of 1 and decaying to $0.001$, whereas the current standard is to keep it constant at $0.001$ (Kazemnejad et al., 2024). They also clip the failure probability ($p_{\text{fail}}$; see Section A.5). Despite these strategies, we observed large sample-dependent weights ($g_N^+$ and $g_N^-$ in Eq. 8), which we countered by directly clipping them to stabilize training.

Unfortunately, none of these interventions enabled the model to solve the `Degree-10-Path-10` dataset.

To investigate further, we applied the method to the `Degree-5-Path-5` dataset which `Dr.GRPO` can effectively solve (see Fig. 7). We believe a major reason is the presence of very high negative gradients. When the failure probability ($p_{\text{fail}}$) is close to 1, the sample-dependent weights become large, and multiplying them by negative log probabilities produces high magnitude negative gradients. This drives the model responses toward degeneracy where it repeats the same set of characters. Figure 4 shows this effect: with a lower KL penalty (0.001), the model's response lengths increase rapidly, and inspection of the outputs confirms degeneracy, while success rates remain zero. Using a strong-to-weak KL penalty (0.1 to 0.001) stabilizes training but does not help solve the hard task.

We think this problem of high negative gradients could be resolved when one uses a capable base model to begin with (low failure probability means lower magnitude sample dependent weights for the negative samples), which is what Chow et al. (2025) work with (also see Fig. 2 where `Best-of-N` aware finetuning is able to solve the `Degree-3-Path-3`, possibly due to relatively lower failure rates initially, compared to `Degree-5-Path-5`).

We believe that the issue of high negative gradients can be mitigated by starting with a capable base model. Such a model has a lower failure probability ($p_{\text{fail}}$ in Eq. 8), which keeps the sample-dependent weights $g_N^+$ and $g_N^-$ (in Eq. 8) within a reasonable range, thus ensuring stable training. As shown in Fig. 2, `Best-of-N` aware finetuning is able to solve the `Degree-3-Path-3` task, likely due to its relatively lower initial failure rates compared to `Degree-5-Path-5`.

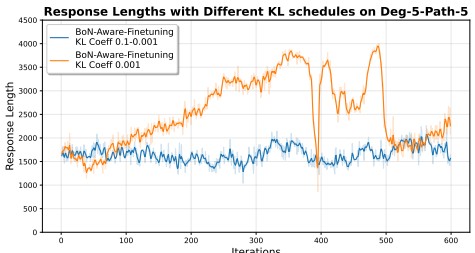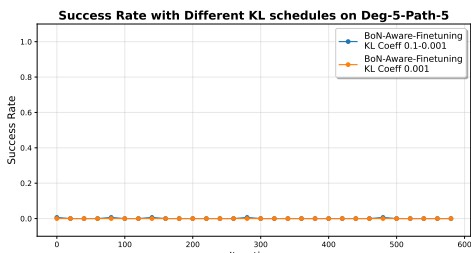

Figure 4: Using a lower KL coefficient, i.e., the standard value of 0.001 in `Best-of-N` aware finetuning, results in unstable training due to large-magnitude negative gradients, causing model responses to degenerate into repeating the same character. In contrast, using a KL schedule as recommended in Chow et al. (2025) (decaying from a strong KL penalty of 0.1 to 0.001) remains stable but fails to learn, as success rates stay at zero (see figure on the right).

> ***Takeaways***
>
> - All baselines **fail under zero outcome rewards** on a simple graph search task, even though some were specifically designed to operate under zero outcome rewards. Baselines that we tested include: **naive RL** Liu et al. (2025), **improving credit assignment** Kazemnejad et al. (2024), **reward shaping** Setlur et al. (2025a), and **`Best-of-N` aware finetuning** Chow et al. (2025).
> - Instantiating `Progress Rewards` is practically challenging, whereas there is a requirement of a *capable* base model to begin with for `VinePPO` and to possibly resolve unstable training of `Best-of-N` aware finetuning.

## 5 ADDING EASY SAMPLES WORKS

As shown in Fig. 1, **mixing a simpler variant of the task**, specifically samples from the `Degree-5-Path-5` dataset, into the original `Degree-10-Path-10` dataset significantly improves performance, allowing the model to solve the task where all baselines had previously failed.

This raises an important question: are all "easy" samples equally effective? In this section, we explore (i) the impact of mixing samples of varying difficulty levels and (ii) how one can, in some cases, bypass the challenge of selecting samples with the appropriate difficulty for a given task.

## 5.1 NOT ALL EASY SAMPLES WORK

Here we consider mixing two other types of easy datasets in equal proportion with the original `Degree-10-Path-10` task. The first easy graph is a `Degree-2-Path-5` graph, where the center node has a degree of 2, and each of the outgoing paths has 5 nodes. The second easy graph is a `Degree-5-Path-2` graph, where the center node has a degree of 5, and each of the outgoing paths has only 2 nodes.

As shown in Fig. 5 (left), in both cases the training rewards saturate around $0.5$. From Fig.5 (right), it is evident that neither of the easier datasets helps in solving the original `Degree-10-Path-10` task. Examining the chain-of-thought traces reveals that: (i) when the `Degree-2-Path-5` task is mixed with the original task, the model often explores only two branches before committing to a final answer. While this strategy works for the easier dataset (where the center node has degree 2), it fails on the `Degree-10-Path-10` task, which requires exploring multiple branches. (ii) In the case of the `Degree-5-Path-2` mixture, instances are solved using a simple adjacency-list lookup, and the model shows no evidence of backtracking on harder graphs. In both settings, the model learns to solve only the easier dataset in the mixture, which explains why the training rewards plateau near $0.5$.

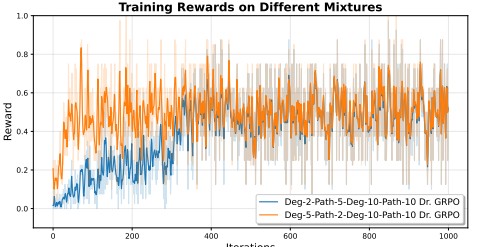 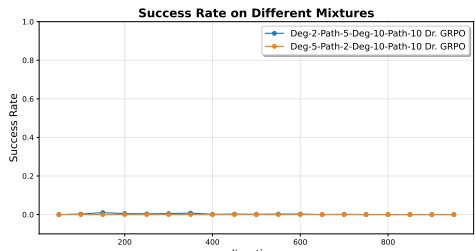

Figure 5: **(Left)**: Rewards that Qwen2.5/Qwen-1.5B-Instruct model obtains while training `Dr.GRPO` on a dataset containing an equal mixture of (i): Degree-5-Path-2 mixed with Degree-10-Path-10, and (ii): Degree-2-Path-5 mixed with Degree-10-Path-10. The training rewards saturate to around $0.5$ in both cases, and in both cases the model learns to solve the easier examples in the mixture. **(Right)**: Success rate on a held-out test set of `Degree-10-Path-10` examples. Both mixtures do not help the model solve the harder `Degree-10-Path-10` task.

## 5.2 MIXING ALL SAMPLES YOU HAVE IS EFFECTIVE

From Fig. 5, it is clear that not all easy samples are equally effective, and it is not obvious in advance what a model will learn from a given dataset. To be useful, samples must have the *right* difficulty, meaning they should be solvable by the model while also encouraging *behaviors* that transfer to the target task. As we saw earlier, adding `Degree-5-Path-5` examples improves performance (Fig. 1), whereas adding much easier examples such as `Degree-5-Path-2` or `Degree-2-Path-5` does not (Fig. 5). For the graph search task, one can probably reason why `Degree-5-Path-2` or `Degree-2-Path-5` is *very easy* and why `Degree-5-Path-5` represents the *right* difficulty. However, for general tasks, it is often difficult to define the *right* difficulty a priori. This requirement of selecting the *right* difficulty makes the approach cumbersome.

However, we find that if one mixes all samples they have of varying difficulty in the training dataset and then train using naive RL (`Dr.GRPO`), the model is able to solve the hard task. To be specific, if one constructs a dataset that combines `Degree-2-Path-5`, `Degree-5-Path-2`, `Degree-5-Path-5`, and `Degree-10-Path-10` examples in equal proportion, and train using `Dr.GRPO`, model learns to solve the `Degree-10-Path-10` task (see Fig. 6). Importantly, the RL algorithm itself remains unchanged, only the data changes .

This means that instead of *choosing* the *right* difficulty samples to aid transfer, one can simply include all available samples of varying difficulty in the training dataset, making the approach much simpler. Importantly, the model still learns the *right* behavior on its own (Gandhi et al., 2025), likely from the samples of appropriate difficulty that facilitate transfer to the hard task. This provides a **practical recipe** for an RL practitioner.

Thus, if the base model cannot solve the task initially, adding samples of varying difficulty can help it succeed. Moreover, even when the base model can already solve the task, easy-to-hard curricula that is implicit in the dataset can accelerate convergence (Stojanovski et al., 2025).

### 5.3 WHY DOES THIS WORK?

We believe there may be an interesting connection to skill learning here (Eysenbach, 2025). Easier samples allow the model to acquire certain skills (or *correlated actions*) from outcome rewards alone. These learned skills can then transfer to more difficult tasks that the base model could not originally solve. Another way to view this is that *correlated actions* (or skills) simplify the search problem during RL: the model now searches in the space of skills rather than in the raw space of tokens (which is much larger). In other words, including examples of the appropriate difficulty in the training mixture helps reduce the effective action space when doing RL, making it easier for the model to sample a successful rollout. For example, one useful skill for this task is traversing down a branch to a leaf node without hallucinating and then backtracking. Another is systematically exploring all branches one by one. That said, these are anthropomorphized descriptions; ultimately, what counts as a "skill" is determined by the RL optimization process.

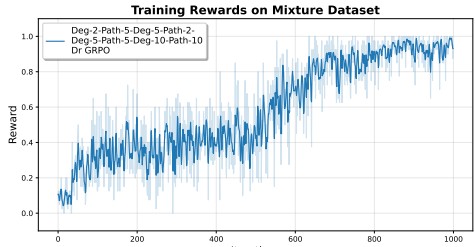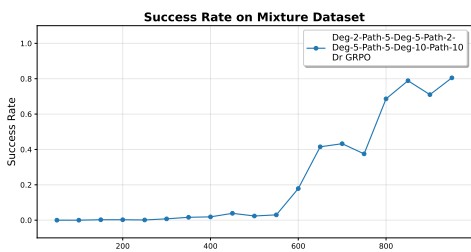

Figure 6: **(Left)**: Rewards obtained while training on an equal-proportion mixture of `Deg-2-Path-5`, `Deg-5-Path-2`, `Deg-5-Path-5`, and `Deg-10-Path-10` using `Dr.GRPO`. During training, the model learns to solve all elements in the mixture, with rewards reaching 1.0. **(Right)**: Success rate on a held-out test set of `Degree-10-Path-10` examples. The model trained on the mixture dataset begins solving the harder task.

> ***Takeaways***
>
> - Simply mixing easier samples in the training dataset and doing naive RL helps the model solve the original hard task, even though the base model was unable to do so initially.
>
> - Not all easy samples work: adding very easy samples in the training dataset does not help. Adding the *right* difficulty matters!
>
> - Final practical recipe: Instead of making a choice for what's the *right* difficulty, add all samples of varying difficulty that you have. The base model still learns the *right* actions from the *right* difficulty samples that transfer to the hard task.

## 6 RELATED WORK

**Curriculum learning and data difficulty control:** Recent work highlights the importance of curricula in developing reasoning abilities for large language models. Reasoning Gym (Stojanovski et al., 2025) introduces parameterized reasoning tasks of varying difficulty and shows that easy-to-hard training accelerates convergence. Building on this idea, E3 (Explore–Exploit–Extrapolate) (Setlur et al., 2025b) jointly varies task difficulty and token budgets to gradually expand in-context

reasoning. WISDOM (Qiu et al., 2025) takes a similar approach by generating chain-of-thought data of increasing difficulty to expose models to progressively harder math problems, while SEC (Chen et al., 2025b) adapts problem selection using a multi-armed bandit based on the model's learning progress. Together, these studies demonstrate that carefully controlling what data a model sees and when it sees it is a powerful mechanism for scaling reasoning in LLMs. However, they still assume that the base model begins with some initial level of capability.

**Self-play and self-generated supervision:** Several methods leverage self-play or model-generated data to provide supervision in low-reward scenarios. Cheng et al. (2024) show that an adversarial two-player game played by copies of an LLM can improve its reasoning through RL on game outcomes. Wang et al. (2025) introduce a Critic-Discernment Game (CDG), in which a prover model is alternately challenged by helpful and misleading critics, training it to maintain correct answers despite adversarial feedback. More recently, methods such as Self-Questioning LMs (Chen et al., 2025a) allow models to generate their own subtasks and verifiers, creating a self-curriculum without relying on external datasets. These approaches highlight the use of self-play loops and self-generated supervision to bootstrap training when explicit reward signals are sparse or absent.

**Reward shaping and credit assignment in sparse RL:** A complementary line of work addresses the sparse reward problem by shaping rewards or refining credit assignment. Classic methods such as Hindsight Experience Replay (Andrychowicz et al., 2017) demonstrate that relabeling failed trajectories as successes for alternate goals allows learning from episodes that would otherwise provide no signal. Building on this idea, recent approaches in LLM reasoning focus on step-level credit assignment, enabling models to extract useful learning signals even from failed trajectories. For example, VinePPO (Kazemnejad et al., 2024) uses Monte Carlo rollouts to compute step-level advantages, yielding stable improvements over vanilla PPO (Schulman et al., 2017), while Rewarding Progress (Setlur et al., 2025a) trains $Q$-functions to score intermediate reasoning steps based on their progress towards the final answer, substantially improving sample efficiency. OREAL (Lyu et al., 2025) takes an orthogonal approach by combining best-of-N outcome supervision (Chow et al., 2025) with token-level reward models to propagate credit to critical steps in long reasoning traces. These studies demonstrate that reward shaping and refined credit assignment can mitigate sparse feedback, yet they still assume the presence of some non-zero signal. By contrast, our work focuses on the zero-reward setting, where such techniques fail unless easier instances are provided to bootstrap learning (see Fig. 1).

## 7 LIMITATIONS AND FUTURE WORK

Our analysis focuses on the graph search problem, and it remains an open question to what extent these findings generalize to more complex reasoning domains such as mathematical problem solving or code generation. While we employed a uniform mixture of task difficulties (see Fig. 6), exploring optimal weightings or curricula for easy versus hard tasks could further improve convergence. In addition, our approach relies on a non-zero success rate on simple problems to bootstrap learning. Addressing this cold-start challenge through self-play data generation or more robust pretraining is a promising direction for future work.

## 8 CONCLUSIONS

Our results highlight the critical role of data-centric strategies in RL for reasoning. Algorithmic improvements such as dense rewards, refined credit assignment, and diversity incentives fail when the base model has zero success rates on the task. In contrast, introducing easy instances consistently enables learning by providing a foothold from which the model can bootstrap toward more challenging tasks. We recommend that future evaluations include settings where base models initially fail, as success from these cold-start regimes offers a more robust measure of genuine progress in exploration and reasoning.

## 9 REPRODUCIBILITY STATEMENT

The training objectives for all methods, including `Dr.GRPO`, `VinePPO`, `Progress Rewards`, and `Best-of-N` aware finetuning, are provided in Appendix A. Appendix B lists the hyperparameter used for each method, and Appendix C lists the prompts used in the graph search experiments. To further support replication, we release the codebase containing the implementation for all baselines at https://github.com/anon-zero-rewards/zero-rewards-rl.

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

# A    BASELINE IMPLEMENTATION DETAILS

In this section, we provide background on reinforcement learning (RL) for large language models (LLMs) and describe the objective that different baselines optimize.

## A.1    RL FOR LLMS

Let $\pi_\theta(a \mid s)$ denote a policy parameterized by $\theta$, which defines the probability of taking action $a$ in state $s$. A trajectory of length $T$ is denoted by $\tau = (s_0, a_0, s_1, a_1, \ldots, s_T, a_T)$, and the total reward of a trajectory is $R(\tau)$. In the context of language models, the initial state $s_0$ corresponds to the input question, actions correspond to generated tokens, and subsequent states represent the question along with the partial sequence of answer tokens.

RL training using Reinforce (Williams, 1992) optimizes the following objective:

$$\theta^* = \text{argmax}_\theta \, \mathbb{E}_{x \sim \mathcal{D}} \, \mathbb{E}_{y \sim \pi_\theta(.|x)} R(x, y) \tag{1}$$

Here, $x$ is a question sampled from the distribution of questions $\mathcal{D}$, $y$ is a response sampled from the LLM $\pi_\theta(. \mid x)$, and $R(x, y)$ is the reward for generating response $y$ to question $x$. In our graph search setting, we prompt the LLM to output its final answer inside `\boxed{}`, so the reward function simply extracts the string in `\boxed{}` and compares it to the true path.

## A.2    DR. GRPO (LIU ET AL., 2025)

The objective in Equation 1 typically exhibits high variance. To reduce this variance, recent methods such as GRPO (Shao et al., 2024) Dr. GRPO (Liu et al., 2025) subtract a baseline from the reward, which is computed by generating multiple rollouts for a given question and taking the average reward. In our setting, we perform on-policy training; thus, the importance ratio and advantage clipping terms in the objective from equation 3 in (Liu et al., 2025) vanish, resulting in the following final objective:

$$\nabla_\theta \mathcal{J}(\theta) = \mathbb{E}_{x \sim \mathcal{D}, \{y_i\}_{i=1}^G \sim \pi_\theta(.|x)} \frac{1}{G} \sum_{i=1}^{G} \sum_{t=1}^{|y_i|} \hat{A}_{i,t} \nabla_\theta \log \pi_\theta(y_{i,t} \mid y_{i,<t}) - \beta \, \mathbb{D}_{\text{KL}}(\pi_\theta || \pi_{\text{ref}}) \tag{2}$$

where $\hat{A}_{i,t}$ is computed as $R(x, y_i) - \frac{1}{G} \sum_{j=1}^{j=G} R(x, y_j)$. We set $G = 5$ for all our experiments.

Intuitively, the objective increases the likelihood of tokens that lead to positive outcomes and decreases the likelihood of tokens that do not. Note that this loss formulation weights all steps in the reasoning chain equally, i.e., $\hat{A}_{i,t}$ is independent of $t$.

## A.3    VINEPPO (KAZEMNEJAD ET AL., 2024)

Methods like GRPO and Dr. GRPO weigh all tokens in a reasoning chain equally. However, this may not be optimal, as not all steps in a reasoning chain contribute equally to solving a problem. For example, identifying lemmas to use in proving a theorem may be more important than generating tokens that are included merely for grammatical correctness. VinePPO addresses this problem by performing credit assignment to weigh more important steps higher and less important steps lower. It does so by computing step level advantages instead of trajectory level advantages.

Given a question $x \sim \mathcal{D}$, and a reasoning chain $y \sim \pi_\theta(. \mid x)$, VinePPO divides the reasoning chain $y$ into steps or chunks $y_{c_1}, y_{c_2} \ldots y_{c_n}$, and finally the gradient of the objective is computed as

$$\nabla_\theta \mathcal{J}(\theta) = \sum_{i=1}^{n} \hat{A}_{y_{c_i}}^{\pi_\theta} \sum_{t=s_{c_i}}^{e_{c_i}} \nabla_\theta \log \pi_\theta(y_t \mid y_{<t}, x) \tag{3}$$

where $\hat{A}^{\pi_\theta}_{y_{c_i}}$ denotes the advantage of step $y_{c_i}$, with $s_{c_i}$ and $e_{c_i}$ representing the start and end indices of the $i^{\text{th}}$ chunk, respectively. The advantage at each step is estimated using Monte Carlo rollouts from the state before and after that step. In other words, given a trajectory $y = y_{c_1} \oplus y_{c_2} \oplus \ldots \oplus y_{c_n}$ where $\oplus$ denotes the concatenation operation $\hat{A}^{\pi_\theta}_{y_{c_i}}$ is computed as follows:

$$\hat{A}^{\pi_\theta}_{y_{c_i}} = V^{\pi_\theta}(\oplus_{j=1}^{j=i} y_j) - V^{\pi_\theta}(\oplus_{j=1}^{j=i-1} y_j) \tag{4}$$

where $V^{\pi_\theta}(s)$ is computed by taking the average reward recieved from $K$ Monte Carlo rollouts. In our experiments we set $K = 3$. Note that we omit the KL penalty term from Equation 3 for the sake of simplicity. For VinePPO, we use the code released by the authors here.

### A.4 PROGRESS REWARDS (SETLUR ET AL., 2025A)

Like the VinePPO objective in Equation 3, progress rewards also aim to compute step-level advantages. However, their objective differs from VinePPO in two key ways: (i) they estimate the advantages in Equation 4 under a policy different from the one being optimized, which they refer to as the prover policy, and (ii) combine the step-level advantages with the trajectory-level reward to form the final objective:

$$\nabla_\theta \mathcal{J}(\theta) = \sum_{i=1}^n (R(y) + \alpha \cdot \hat{A}^\mu_{y_{c_i}}) \sum_{t=s_{c_i}}^{e_{c_i}} \nabla_\theta \log \pi_\theta(y_t \mid y_{<t}, x) \tag{5}$$

where $\mu$ is `Best-of-4`$(\pi_{\text{ref}})$. Similar to equation 4 they compute step advantages under the prover policy $\mu$ as:

$$\hat{A}^\mu_{y_{c_i}} = V^\mu(\oplus_{j=1}^{j=i} y_j) - V^\mu(\oplus_{j=1}^{j=i-1} y_j) \tag{6}$$

In their work, a neural network is trained to approximate $V^{\pi_{\text{ref}}}(s)$ by collecting a dataset of reasoning trajectories sampled from $\pi_{\text{ref}}$. For simplicity, we instead rely on rollouts to compute $V^{\pi_{\text{ref}}}(s)$. With this, we optimize the following objective:

$$\nabla_\theta \mathcal{J}(\theta) = \sum_{i=1}^n (\hat{A}(y) + \alpha \cdot \hat{A}^\mu(y_{c_i})) \sum_{t=s_{c_i}}^{e_{c_i}} \nabla_\theta \log \pi_\theta(y_t \mid y_{<t}, x) \tag{7}$$

Notice the change from Equation 5, where we replace $R(y)$ with $\hat{A}(y)$. We make this substitution because we operate in a group-style setting with access to multiple rollouts for a question. Using $\hat{A}(y)$ instead of $R(y)$ yields a lower-variance estimate of the gradient. $\hat{A}(y)$ is computed as $(R(y) -$ average reward over $G$ rollouts).

### A.5 BEST-OF-N FINETUNING (CHOW ET AL., 2025)

Chow et al. (2025) introduce inference-aware fine-tuning that explicitly optimizes for BoN performance. The hope is optimizing for BoN performance could result in the model sampling *diverse* responses. A key result is Lemma 3, which provides a BoN-aware policy gradient estimator under binary rewards. By introducing asymmetric weighting functions that depend on the probability of failure, the method upweights correct predictions on hard inputs while redistributing mass away from unreliable outputs.

Assuming the reward $R(x, y) \in \{0, 1\}$. Then the gradient of the BoN-aware RL objective (Equation 6) with respect to the model parameters $\theta$ is

$$\mathbb{E}_{x \sim D}\left[\mathbb{E}_{y \sim \pi_{\text{bon}, R=1}}[\nabla_\theta \log \pi_\theta(y \mid x)] \, g_N^+(P_{\text{fail}}(x)) - \mathbb{E}_{y \sim \pi_{\text{bon}, R=0}}[\nabla_\theta \log \pi_\theta(y \mid x)] \, g_N^-(P_{\text{fail}}(x))\right],$$
$$\tag{8}$$

where the sample-dependent weights are

$$g_N^+(p) = \frac{N\,p^{N-1}}{1-p^N}, \qquad g_N^-(p) = \frac{N\,(1-p^{N-1})}{1-p^N}.$$

## B HYPERPARAMETERS

Table 1 lists the hyperparameters that are common across all methods. Since we do on-policy training parameters such as the clipping ratio ($\epsilon$) and `ppo-epochs` are not applicable.

| Hyperparameter | Value |
|---|---|
| Learning Rate | $1 \times 10^{-6}$ |
| Effective Batch Size | 32 |
| Number of rollouts per data point | 5 |
| Max Prompt Length | 1024 |
| Max Response Length | 4096 |
| KL Coefficient ($\beta$) | $1 \times 10^{-3}$ |
| Temperature | 0.6 |
| Top p | 0.999 |
| Micro Batch Size | 4 |
| Discount Factor ($\gamma$) | 1 |
| Base Model | Qwen2.5/Qwen-1.5B-Instruct |

Table 1: Common hyper-parameters used in our experiments.

### B.1 VINEPPO

We estimate the value of an intermediate state, $V^{\pi_\theta}(s)$, using three rollouts from the current policy.

### B.2 REWARDING PROGRESS

We estimate the value of an intermediate state, $V^{\pi_{\text{ref}}}(s)$, using three rollouts from the reference policy. The parameter $\alpha$ in Equation 7 is set to 5, following the recommendation in Setlur et al. (2025a).

### B.3 BEST-OF-N AWARE FINETUNING

For $N$ used in Best-of-N, we use $N = 8$. We clip the sample dependent weights to $[-3, 3]$. We decay the KL coefficient from 0.1 to 0.001.

## C PROMPTS

For our experiments, we prompt the Qwen2.5/Qwen-1.5B-Instruct model in the following manner

---

### Prompt for path finding problem

Given a bi-directional graph in the form of space separated edges, output a path from source node to the destination node in the form of comma separated integers.
For this question the graph is 81,252 97,124 285,182 97,285 97,81 124,199
The source node is 97
The destination node is 252
Please reason step by step, and put your final answer within \boxed{}.

---

During training, the string inside \boxed{} is extracted and compared against the ground truth path to assign the reward.

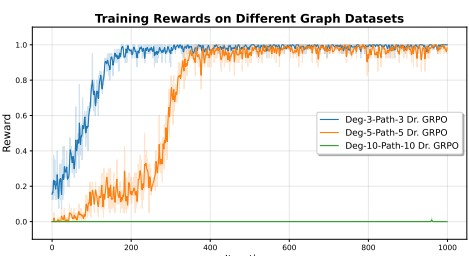
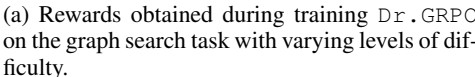
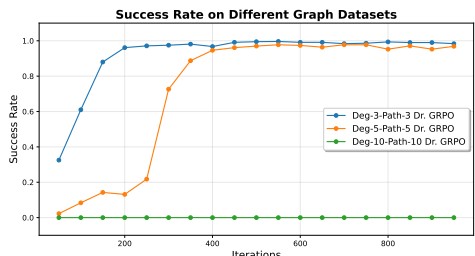

(a) Rewards obtained during training `Dr.GRPO` on the graph search task with varying levels of difficulty.

(b) Success Rate of `Dr.GRPO` on held out test sets of different levels of difficulty. The model manages to solve simpler variants (Degree-3-Path-3 and Degree-5-Path-5), but is unable to solve the harder Degree-10-Path-10 variant.

Figure 7: `Dr.GRPO` is able to solve easier instance of the graph search task.

## D    ADDITIONAL RESULTS

**Easier versions of the task are solvable:**   To rule out any implementation issues and verify that the graph search task is not inherently unsolvable, we experiment with simpler variants of the task, such as the `Degree-3-Path-3` and `Degree-5-Path-5` graphs. As shown in Figs. 7a and 7b, RL training using outcome rewards is effective on these simpler variants, but remains ineffective on the harder `Degree-10-Path-10` dataset.

