# OpenReview forum: "What Can You Do When You Have Zero Rewards During RL?"
_ICLR.cc/2026/Conference — ICLR 2026 Conference Withdrawn Submission_

### Official Review · Reviewer_S9eA · 2025-10-31

**Soundness:** 3
**Presentation:** 2
**Contribution:** 3
**Rating:** 4
**Confidence:** 3

**Summary:**

This work investigates the zero-reward barrier in reinforcement learning for large language models. These are scenarios where the base model cannot generate any correct solutions during training, causing learning to stall. Using a graph search task from Bachmann and Nagarajan (2024), the authors evaluate recent RL methods designed for sparse rewards: VinePPO (credit assignment), Rewarding Progress (reward shaping), and Best-of-N aware finetuning (diversity incentives). The authors implement baselines (providing code) and provide detailed failure analyses. They release their implementations to support further research.

The strengths of the work are as follows. First, the zero-reward barrier is a fundamental challenge limiting RL scalability for LLMs. Second, the authors implement and rigorously test multiple recent methods specifically designed for sparse rewards. Third, there is a thorough failure mode analysis on dense rewards. Finally, the work is reproducible and the authors provide complete code.

The weaknesses of the work are as follows. First, there is a limited task diversity since the work only focuses on graph search tasks. It would be great to see some analysis on if this transfers to math, coding, etc. Second, there is only experiments on a 1.5B model. It would be great to see larger models, to see if the trend holds. Finally, some improvements to presentation (e.g. graph readability) could be made in the paper, spacing.

**Strengths:**

The strengths of the work are as follows.
- First, the zero-reward barrier is a fundamental challenge limiting RL scalability for LLMs.
- Second, the authors implement and rigorously test multiple recent methods specifically designed for sparse rewards.
- Third, there is a thorough failure mode analysis on dense rewards.
- Finally, the work is reproducible and the authors provide complete code.

**Weaknesses:**

The weaknesses of the work are as follows.
- First, there is a limited task diversity since the work only focuses on graph search tasks. It would be great to see some analysis on if this transfers to math, coding, etc.
- Second, there is only experiments on a 1.5B model. It would be great to see larger models, to see if the trend holds.
- Finally, some improvements to presentation (e.g. graph readability) could be made in the paper, spacing.

**Questions:**

n/a

---

### Official Review · Reviewer_s9U5 · 2025-10-31

**Soundness:** 1
**Presentation:** 2
**Contribution:** 2
**Rating:** 2
**Confidence:** 5

**Summary:**

The paper explores whether large language models can perform reinforcement learning (RL) style reasoning in a synthetic sparse-reward environment.
The authors design a graph-search task (“Degree-10-Path-10”) and train the LLM through policy-gradient updates using only a few sampled completions per input.
They claim that the failure of training indicates the fundamental infeasibility of applying RL to LLMs for complex reasoning and their curriculum training can solve it.

**Strengths:**

The paper tackles an ambitious question about combining LLMs and RL, which is of high interest to the community.

The authors show enthusiasm in connecting classical RL problems to language-based reasoning, and the overall motivation is understandable.

**Weaknesses:**

1) Insufficient exploration and unrealistic sampling setup.

The experiments rely on only a few rollouts per data point (five completions), under a completely sparse 0/1 reward.
This setup guarantees that almost no successful trajectories are ever observed, so gradients vanish and training fails by design.
The paper then interprets this trivial failure as a general limitation of LLMs, which is misleading.
RL methods typically require thousands to millions of rollouts for such problems; the present configuration is not a valid test of feasibility.

2) Compute–task imbalance.

The graph-search problem (Degree-10-Path-10) explodes combinatorially in difficulty, but the compute and sample budgets remain fixed and minimal.
The failure is thus compute-bound rather than method-bound.
No scaling study is provided to show where learning begins to succeed or how the outcome changes with more computation.
The conclusion that “RL cannot work for LLMs” is not supported without such evidence.

3) Superficial notion of curriculum learning.

The “curriculum” is a manually mixed dataset of easy and hard samples at a fixed ratio.
It is static, non-adaptive, and not connected to the model’s learning progress.
This is better described as data filtering, not curriculum learning.
In modern RL, curricula are dynamic and algorithmic—tasks or rewards evolve based on the agent’s competence.
The paper misses this key distinction and offers no analysis of why or when their fixed mixture helps.
Moreover, the current setup risks overfitting to “easy” samples and does not test generalization to unseen or multi-solution cases.
The authors also have no idea about the works of the meta-learning and curriculum learning community from RL.
I would expect more holistic solutions rather than simply picking samples.
When the algorithm try more reasoning tasks, do you also manually give easier samples?

4) Missing comparisons and baselines.

No results from standard RL algorithms are shown for the same environment.
Without knowing that a baseline agent can solve the task, the “failure” of the LLM provides little information.
The paper also lacks ablations on rollout number, reward shaping, or task scaling.

5) Overstated conclusions.

The authors generalize their negative results to claim that RL on LLMs is infeasible, but the evidence only shows that the current configuration is under-explored and under-resourced.
The conclusions are therefore not justified by the experiments.

6) Clarity and structure issues.

The writing is fragmented, with vague definitions, missing implementation details, and inconsistent terminology (e.g., “rollout,” “curriculum,” “episode”).
The overall presentation makes it difficult to follow the core argument.

**Questions:**

What happens if the number of rollouts or compute budget is scaled up by one or two orders of magnitude? Does performance improve at all?

Could the authors test smaller or simpler tasks (e.g., lower degree/path length) to see when learning actually starts?

Can they design a dynamic curriculum that adjusts difficulty automatically instead of fixing ratios?

How do classical RL baselines and other coldstart RL algorithms perform on the same environment? Without comparison, it is unclear whether the task itself is solvable.

---

### Official Review · Reviewer_ufeb · 2025-10-31

**Soundness:** 2
**Presentation:** 2
**Contribution:** 2
**Rating:** 2
**Confidence:** 4

**Summary:**

Paper studies the “zero-reward barrier” in RL post-training for LLMs: when a base model never samples a correct answer, outcome-based RL stalls. Using a controllable graph search task (Degree-10-Path-10), the authors evaluate Dr.GRPO, VinePPO, Rewarding Progress, and BoN-aware finetuning and report persistent zero success.

Core result: a data-centric fix—mixing in easier instances (not too easy) enables learning on the hard task without changing the RL algorithm; equal-mixing all difficulties also works.

**Strengths:**

All baselines flatline at zero success on Deg-10-P10, and the paper convincingly rules out trivial explanations by showing Deg-3-P3 is solvable.It offers a simple fix: include easier instances (Deg-5-P5) or mix difficulties, which reliably unlocks learning under outcome rewards. They explain why step-credit still gives no signal when success=0 and why BoN is unstable without strong KL/clipping.

**Weaknesses:**

1. The paper focuses on a single controllable toy task. While that’s fine for isolating dynamics, I’d expect the core insight (“mix difficulties to break zero-reward stalls”) to be validated on more realistic benchmarks. There are no results on real-world-style tasks, so external validity is unclear.
2. Running everything on one small, single family/model caps the claim. The zero-reward barrier and the benefit of mixing difficulties likely depend on base competence and instruction-tuning/tokenizer quirks; a larger Qwen or a different family (e.g., Llama/Mistral) might start with non-zero success on the hard split, changing both learning curves and the size of the effect.

**Questions:**

1. Why did the authors only single task?
2. Curious to know what authors think when the model size is larger and if the claims still hold.
3. Why did the authors only consider single model in this paper to validate the claims?

---

### Official Review · Reviewer_YCaD · 2025-11-11

**Soundness:** 2
**Presentation:** 2
**Contribution:** 1
**Rating:** 2
**Confidence:** 3

**Summary:**

This paper explores the "zero reward barrier" in reinforcement learning with LLMs. This occurs when the model is incapable of generating outputs that yield non-zero rewards, thus has no meaningful gradient for updating the policy. The paper proposes a simple curriculum-style solution: inject easier problems to the training set. This is studied in the context of a graph search problem, and performance is compared to baseline approaches.

**Strengths:**

* The identified problem is an important (if well-known) one. This failure mode of RL is especially important in the context of LLM training and the increasing application of them towards difficult tasks. While I think this paper would benefit from improved and more thorough analysis, it doesn't change the fact that the problem itself is important to the community.

* The discussion on why existing methods fail in the sparse reward setting are helpful (Sec. 4). While some of these failure modes are somewhat obvious, the exploration of the prover for Progress Rewards and degeneracy in Best-of-N provide some nice insights.

* Clean implementations of methods are always greatly appreciated for the sake of reproducibility, and the community would benefit from the ones put forward here.

**Weaknesses:**

* At risk of being overly reductive, this just seems like a curriculum. The idea that a lack of meaningful rewards complicates learning (which is especially common in sparse reward settings) has long been known in reinforcement learning. There are many common approaches to help mitigate this: curriculum, intelligent exploration, transfer learning, reward shaping, prioritized/hindsight experience (if off policy) etc. While some of these are indeed mentioned in the paper itself, it doesn't change the fact that this isn't by any means a novel finding in the context of RL. This is particularly apparent in Sec. 5.3, as this reasoning is exactly the purpose behind developing a training curriculum (as well as other replay-based techniques).

* The above issue isn't necessarily a fatal flaw in the paper if there is a suitably rigorous analysis with interesting findings. And while I do appreciate the analysis of existing technqiues in Sec. 4, ultimately it is applied only to a graph search problem and a single LLM (Qwen 2.5 1.5B). While I agree that this is a useful problem, it does mean that the curriculum that is being implicitly introduced is only studied in the context of one problem and one model.

* I like the discussion on what is the "right" difficulty of curriculum, e.g. what samples should be chosen (5.1/5.2). This has always been a challenging problem in curriculum design in RL and I think it is very useful to explore in the context of LLM training. However, there is no satisfying answer to this problem in the paper other than simply adding samples of all difficulties, which makes it difficult to apply more broadly to other tasks.

**Questions:**

1. How many "easy" examples are required to bootstrap learning? Can we use just a small amount and sample them with a higher frequency to kickstart learning?

2. Can the need for "easy" examples be avoided by providing some teacher-based exploration? E.g. use the answer/output from a stronger model/human when early in training.

---

### Note · Authors · 2025-11-14

I have read and agree with the venue's withdrawal policy on behalf of myself and my co-authors.